# Lessons Learned from a Distributed RF-EMF Sensor Network

**DOI:** 10.3390/s22051715

**Published:** 2022-02-22

**Authors:** Sam Aerts, Günter Vermeeren, Matthias Van den Bossche, Reza Aminzadeh, Leen Verloock, Arno Thielens, Philip Leroux, Johan Bergs, Bart Braem, Astrid Philippron, Luc Martens, Wout Joseph

**Affiliations:** 1WAVES, Ghent University/imec, Technologiepark-Zwijnaarde 126, 9052 Ghent, Belgium; gunter.vermeeren@ugent.be (G.V.); matthias.vandenbossche@ugent.be (M.V.d.B.); leen.verloock@ugent.be (L.V.); arno.thielens@ugent.be (A.T.); luc1.martens@ugent.be (L.M.); wout.joseph@ugent.be (W.J.); 2Unitron NV-Unitron Connect, Frankrijklaan 27, 8970 Poperinge, Belgium; reza.aminzadeh@unitrongroup.com; 3IDLab, Ghent University/imec, Technologiepark-Zwijnaarde 126, 9052 Ghent, Belgium; philip.leroux@ugent.be; 4IDLab, University of Antwerp/imec, Sint-Pietersvliet 7, 2000 Antwerp, Belgium; johan.bergs@uantwerpen.be (J.B.); bart.braem@uantwerpen.be (B.B.); 5ASTRID, 1000 Brussels, Belgium; astrid.philippron@astrid.be

**Keywords:** radiofrequency electromagnetic fields (RF-EMF), spatiotemporal exposure assessment, sensor node, distributed sensor network

## Abstract

In an increasingly wireless world, spatiotemporal monitoring of the exposure to environmental radiofrequency (RF) electromagnetic fields (EMF) is crucial to appease public uncertainty and anxiety about RF-EMF. However, although the advent of smart city infrastructures allows for dense networks of distributed sensors, the costs of accurate RF sensors remain high, and dedicated RF monitoring networks remain rare. This paper describes a comprehensive study comprising the design of a low-cost RF-EMF sensor node capable of monitoring four frequency bands used by wireless telecommunications with an unparalleled temporal resolution, its application in a small-scale distributed sensor network consisting of both fixed (on building façades) and mobile sensor nodes (on postal vans), and the subsequent analysis of over a year of data between January 2019 and May 2020, during which slightly less than 10 million samples were collected. From the fixed nodes’ results, the potential errors were determined that are induced when sampling at lower speeds (e.g., one sample per 15 min) and measuring for shorter periods of time (e.g., a few weeks), as well as an adequate resolution (30 min) for diurnal and weekly temporal profiles which sufficiently preserves short-term variations. Furthermore, based on the correlation between the sensors, an adequate density of 100 sensor nodes per km^2^ was deduced for future networks. Finally, the mobile sensor nodes were used to identify potential RF-EMF exposure hotspots in a previously unattainable area of more than 60 km^2^. In summary, through the analysis of a small number of RF-EMF sensor nodes (both fixed and mobile) in an urban area, this study offers invaluable insights applicable to future designs and deployments of distributed RF-EMF sensor networks.

## 1. Introduction

Wireless communications infrastructure is currently blooming, particularly in urban environments. The fifth generation of telecommunications networks (5G) is introducing new frequencies—between 700 MHz and 60 GHz—to the current range of environmental radiofrequency (RF) electromagnetic fields (EMF), and growing smart city networks are employing RF-EMF for wireless communications between geographically distributed sensors, embracing the Internet of Things (IoT) [1]. These additions to the RF environment continue to alter our everyday exposure to environmental RF-EMF. In order to appease public uncertainty and anxiety about RF-EMF and overcome associated delays in the rollout of telecommunications infrastructure [2], it is crucial to continuously monitor the environmental levels of RF exposure, not only in space, but also in time. In addition, spatiotemporal assessment of RF-EMF is of general interest to epidemiologists and governmental agencies dealing with environmental issues [3,4].

There are, or have been, RF-EMF monitoring networks in European countries such as France, Greece [5,6], Italy [7], Malta, and Portugal [8], in which a number of high-end, relatively expensive monitoring systems were spread over a large geographical area. Furthermore, mobile RF monitoring has been performed with car-mounted measurement devices [9,10,11]. However, in recent years, the advent of IoT infrastructure has made it feasible to deploy distributed (wireless) networks of cheaper, low-complexity sensing devices to monitor various environmental parameters, such as temperature, humidity, and air quality, autonomously over a long period. Smart Cities such as the SmartSantander IoT platform in the city of Santander, Spain, and the City-of-Things platform in Antwerp, Belgium, can straightforwardly incorporate a network of RF sensors in order to provide effortless public access to up-to-date information on the current RF-EMF exposure [12,13]. Nonetheless, the large amount of densely-distributed RF-EMF sensors that is required to realize accurate spatiotemporal exposure maps on a local scale remains a serious economic and logistic challenge, which makes the availability of a cheap RF-EMF sensor no less than essential [12].

This paper describes the design and the deployment of a distributed network of low-cost RF sensor nodes in Antwerp, Belgium. The new RF sensor node was designed with off-the-shelf components and was able to sample four frequency bands at high temporal resolution. The deployed distributed sensor network consisted of ten fixed sensor nodes scattered within a small geographical area in the city centre of Antwerp and four mobile sensor nodes installed on vans of the local postal services. All fourteen nodes were calibrated in an anechoic room, and after deployment, the fixed sensor nodes were further validated in situ with specialized RF-EMF measurement equipment. The sensor nodes in the fixed network boasted an unrivalled temporal resolution, while the mobile sensor network covered an extensive geographical area of a size which had been previously unattainable. The fixed sensor nodes’ high temporal resolution was further used to calculate the potentially induced errors in spatiotemporal RF exposure assessment with lower sampling rates and/or fewer days of measurements. From this, an adequate sampling rate for future fixed network deployments as well as an adequate resolution to assess the short-term (diurnal/weekly) variability was deduced. Furthermore, through analysis of the temporal correlation between the sensors and a subsequent clustering, the covered area was divided into regions of similar temporal behaviour, based on which an adequate sensor density was obtained. Finally, from the mobile measurements, potential hotspots of RF-EMF exposure were identified within an area of roughly 60 km^2^ around Antwerp.

In summary, through the analysis of a small distributed number of such sensors (both fixed and mobile) in an urban area, this paper offers invaluable insights applicable to future designs and deployments of distributed RF-EMF sensor networks.

## 2. Materials and Methods

### 2.1. RF-EMF Sensor Node

For this study, an RF-EMF sensor node (further also denoted as ‘RF sensor node’; Figure 1) was designed to be able to measure the RF-EMF exposure in four frequency bands at a rate of at least one sample per band per second (Table 1). To minimize the costs, all of the components were off the shelf, and no specific component design was performed. The frequency bands considered in this study are used in Belgium by second to fourth generation (2G–4G) cellular or mobile telecommunications technologies—Global System for Mobile Communications (GSM; 900 and 1800 MHz), Universal Mobile Telecommunications System (UMTS; 900 and 2100 MHz), and Long-Term Evolution (LTE; 1800 MHz)—as well as by Wireless Fidelity (Wi-Fi; 2400 MHz). The three cellular frequency bands contain only downlink communications, i.e., from the cellular base station to the user, whereas the Wi-Fi band contains bidirectional communications. More specifications of the designed sensor are listed in Table 1.

In order to increase the isolation between the considered frequency bands, the sensor node contains four dedicated narrowband half-wavelength monopole antennas (black stub antennas in Figure 1). Each stub antenna is connected to a frequency-band specific transmission line on a custom printed circuit board (PCB) using a sub-miniature A (SMA) connector, which then leads to a surface acoustic wave (SAW) filter tuned to the respective frequency band. After filtering the signal, a true-root-mean-square (tRMS) RF detector (type HMC1020LP4E) measures its power (Figure 1b). This detector has a dynamic range limited at 70 dB, with a lower detection level of 65 dBm, and a programmable integration bandwidth within the frequency range 0 Hz to 3.9 GHz. The detector provides an analogue output, which is then converted to a 12 bit value using an internal analogue-to-digital convertor (ADC; type ADS1015) at a 5 ms internal sampling rate. The four individual input channels of the ADC (Figure 1b) permit fast and easy switching between the four RF tracks and ensure fast sequential measurements per band. The digital values are then processed in an energy-efficient microcontroller (type ATSAMD cortex M0) to obtain the linear average of the detected RF power within each second (averaged over 200 samples) which is communicated to the backbone via a universal asynchronous receiver–transmitter (UART). The final exposure value (in power density *S* or electric field strength *E*) is calculated via the effective antenna aperture (AA) determined during the calibration (see Appendix A) using S=P/AA and $E=\sqrt{120\pi \;S}$.

The communication protocol and power supply between the smart city gateway and the RF sensor was via a universal serial bus (USB) set up as a virtual communications port (Figure 1b). This communication is one-way in the direction of the gateway.

Because of the separate antennas, the RF sensor node is larger than the exposimeter designed for the SmartSantander IoT platform [14]. However, in contrast to body-worn exposimeters, in a smart city context—where the node is attached to a building, placed on top of a car, etc.—the sensor node’s size is less restrictive.

The planned sensitivity of the RF sensor was 0.02 V/m or 1 µW/m^2^. Given the dynamic range of 70 dB, the upper limit would be 63 V/m (10 W/m^2^), which is higher than the ICNIRP reference level for the general public up to 2400 MHz (i.e., 61 V/m) [15]. The sensitivity reached was further assessed per RF sensor during calibration (Appendix A) and an in situ validation (Appendix B).

Finally, outside the pass bands (Table 1), the attenuation of the SAW filters is typically between 17 and 24 dB relative to the passband insertion loss.

### 2.2. Deployment of Fixed RF-EMF Sensor Network

In January 2019, ten RF sensor nodes (with IDs EMF08–EMF13, EMF17–EMF19, and EMF21) were mounted on building façades, at a height of 4.0–7.5 m, within an area of 0.11 km^2^ in the city centre of Antwerp, Belgium (the IMEC City of Things ‘Smart Zone’, Available online: https://antwerpsmartzone.be/en/ (accessed on 22 December 2021); Figure 2a). These fixed sensor nodes were connected to the mains and to a gateway with wired internet access (Figure 2b).

The fixed sensor nodes were set to sample the four considered frequency bands (900, 1800, 2100, and 2400 MHz; Table 1) once per second (sampling time Δ*s* = 1 s), and the samples were collected in a central database.

### 2.3. Deployment of Mobile RF-EMF Sensor Network

In November 2018, four RF sensor nodes were placed on top of postal vans based in Antwerp (available online: www.bpost.be (accessed on 22 December 2021)). These mobile sensor nodes were connected to a long-range wide area network (LoRaWAN) gateway and to the vehicle’s onboard direct current (DC) power. Their sampling time Δ*s* was set to 30 s in order to lower the burden on the LoRa network. The postal vans did not have a fixed schedule or route nor a fixed speed. Furthermore, the effective capture and transmission of sampling data depended on the vehicle’s driver to plug in the node.

Between 16 November 2018 and 11 December 2019 (390 days), the mobile nodes spanned an area of roughly 35.4 by 51.6 km (roughly 6% of Belgium; Figure 3a). However, 95% of the measurement locations were found in an area of 6.6 by 10.5 km (3.8% of the total area; red rectangle in Figure 3). This area was delineated by taking the 98% intervals (i.e., between the 1st and 99th percentiles) of the samples’ (x, y)-coordinates. Moreover, because GPS coordinates have an inherent uncertainty, they were mapped (or ‘snapped’) to the streets using the algorithm described in [16] (example shown in Figure 4).

Unfortunately, there were almost no mobile measurements in the area covered by the fixed sensors (Figure 3b).

## 3. Results

The results of this study are divided into three parts. In the first part, the temporal behaviour of the RF-EMF measured by the sensors in the fixed network is analysed. Based on this, the second part explores in detail three design parameters of a future sensor network: the sampling speed of the sensor, the temporal resolution of the output data, and the spatial density of the sensor network. Finally, in the last part, the search for RF-EMF exposure hotspots in an extensive geographical area using a mobile sensor network is examined.

### 3.1. Temporal Variability of RF-EMF

In order to reduce the computational load during post-processing, the fixed sensor data were first resampled to a Δ*s* of 30 s by taking the median per 30 s period. This way, between 29 January 2019 and 18 May 2020 (475 days), the fixed sensors captured, on average, 809 k 30 s samples per frequency band, corresponding to an average temporal coverage (i.e., the relative on-time of the sensor) of 59%, with a range of 11% to 96% (Table 2). Periods of missing data (i.e., loss in temporal coverage) were caused by either connectivity loss or power issues. Nevertheless, each sensor managed to collect more than 145 k 30 s samples, which was much more than the maximum number of samples captured by any one sensor during a similar measuring period in the state-of-the-art SmartSantander RF monitoring network (i.e., 94 k samples in 434 days, albeit at a Δ*s* of 5–10 min) [13].

In general, the 900 MHz band constituted the highest contribution (of the four assessed bands) to the total outdoor environmental RF-EMF exposure (as was also previously observed in RF-EMF exposure assessment studies in Belgium [17,18], although the difference with the 1800 MHz band was small (Table 2). Furthermore, the outdoor Wi-Fi levels (measured in the 2400 MHz band) were often too low to be measured.

Figure 5 then shows the weekly averaged electric field strength measured by the various sensors in the 900 MHz band. Although there are indications of seasonal variations (e.g., summer vs. winter months, see EMF08, EMF13, and EMF18 in Figure 5), the measurements did not continue long enough for a thorough analysis.

Figure 6 further shows the maximum and 90% ‘temporal variabilities’, i.e., the logarithmic difference between the maximum and minimum, *V_max_*, or the 95th and 5th percentiles, *V*_90_, of the measurement samples [13], of the fixed sensors’ samples. The variability was the highest in the 1800 MHz frequency band (with the median about 4 dB higher than the next highest, the 2100 MHz band), used by GSM and presumably predominantly LTE, and lowest in the 2400 MHz frequency band (*V*_90_ under 3 dB). The results were similar to those found in the SmartSantander IoT sensor network [13], despite the 15–30 times slower sampling rate in Santander (Δ*s* = 5–10 min).

All measured exposure levels were low (<0.05%) compared to the ICNIRP reference levels for the general public (e.g., 41 V/m for 900 MHz) [15].

Then, as in [13], temporal profiles (TPs) of the RF exposure were made, which express the average variation of the exposure, *η*, over a certain time interval (in this case a week and a day). To determine *η*, the power density samples were first normalized to the average of the considered time interval and then averaged over the entire measuring period. The goal of a TP is to improve (spatiotemporal) interpolation of the RF-EMF exposure (e.g., [13]) by rescaling samples taken at different times of the day or week to the same time instance. For example, a TP can be applied in measurement campaigns where only a single or a few measurements have been performed at each assessment location, each at a different time of day. In this study, both diurnal and weekly temporal profiles (denoted as TP_D_ and TP_W_, respectively) were created. In contrast to the fixed temporal resolution (denoted as ‘Δ*t*’) of 1 h in [13], Δ*t* of 30 s (i.e., the sampling time of both the fixed (after resampling) and mobile nodes in this study) up to 4 h were investigated here.

Diurnal and weekly temporal profiles with Δ*t* = 30 s for the four considered frequency bands are shown in Figure 7 and Figure 8, respectively.

The largest temporal variations in exposure were found for the 1800 MHz band, then for 2100 MHz, 900 MHz, and lastly, small variations were observed for 2400 MHz. At a sampling time Δ*s* of 30 s and subsequent TP resolution of 30 s, fast-changing phenomena were noticed in the diurnal exposure: the sudden increases at 04H and 05H for 1800 MHz (Figure 7b), at 04H, 05H, and 06H for 2100 MHz (Figure 7c), and (barely observable) at 05H and 06H in 900 MHz (Figure 7a); and the decreases at 21H, 22H, 23H, and 00H20 for 2100 MHz (Figure 7c), and (barely observable) drops at 22H and 23H for 900 MHz (Figure 7a). These phenomena indicate discrete changes in the output power of some base station radios in or around the fixed sensor network area. Due to the lower sampling speed of the sensors (5–10 min) and temporal resolution (1 h) in the SmartSantander network, they could not be observed there (if they were present at all) [13].

Furthermore, there were clear differences in RF exposure in the 1800 and 2100 MHz bands between the days of the week, especially between Saturday, Sunday, and the weekdays (Figure 8). For these bands, the highest relative exposure was observed on Saturday morning up to noon (Figure 8b,c). For 2400 MHz, there were numerous peaks in relative exposure measured by EMF09 (and a few by EMF11) (Figure 8d), but they were solitary bursts and were thus not as strongly present in the daily profile (Figure 7d).

### 3.2. Lessons for Future RF-EMF Sensor Networks

The sensor nodes in this study have a high external sampling speed, which is a rare feature, due to frequent design constraints on power, storage, and/or bandwidth for transmission of the data (such as on the SmartSantander platform [12,13]). Here, the high temporal resolution is exploited to assess the potentially induced errors by having to resort to a lower temporal resolution of the reported data and/or to lower sample rates.

First, as there is a lot of jitter in the TPs with a Δ*t* of 30 s (Figure 7 and Figure 8), they were subsequently smoothed, using lower resolutions (longer Δ*t*) instead. As an example, diurnal temporal profiles for the 1800 MHz band with four other Δ*t* (5 min, 30 min, 1 h, and 3 h) are shown in Figure 9.

From Figure 9, it becomes clear that with Δ*t* = 30 min, some of the sudden, discrete changes in exposure remain visible (compare Figure 9b to Figure 7b), while a resolution Δ*t* of 4 h provides only a vague idea of the diurnal variability (compare Figure 9d to Figure 7b). At Δ*t* = 1 h, they remain visible for only one of the nodes (EMF08), which showed the highest jump in relative exposure.

Using a lower resolution clearly introduces an additional uncertainty to the TP. Figure 10 shows the (sensor-averaged) 95th percentiles of the (absolute) relative errors (denoted as ‘Δ_t,95′_) that are potentially induced when using different resolutions Δ*t*, compared to the baseline TPs with Δ*t* = 30 s (Figure 7 and Figure 8). In Figure 10, a resolution Δ*t* of 24 h (1440 min) was added to show the error when not using a TP, but only the daily averages.

Both diurnal (Figure 10a) and weekly figures (Figure 10b) follow the same trends: the largest errors were found for 1800 MHz, due to the high temporal variability in this band (Figure 6, Figure 7 and Figure 8), while the errors for 900 and 2400 MHz remain below ~10% no matter the resolution, except when using only daily averages for TP_w_ (Figure 10b), owing to their low or even negligible variability.

In the assumption that lower sampling speeds are generally more useful, for example when using battery-powered sensors, it can be concluded from Figure 10 that Δ*t* = 30–60 min—for which Δ_t,95_ remains below 33% (1.2 dB)—would be a fair trade-off between accuracy and speed. In practice, this aligns with the ICNIRP (2020) guidelines that prescribe an averaging time of 30 min for environmental RF exposure assessment of the general public.

Then, TP_D_ and TP_W_ with Δ*t* = 60 min and based on an output sampling time Δ*s* of 30 s (e.g., Figure 9c) were used as a baseline to assess the potential errors induced by a lower sampling speed. To this goal, Δ*s* of 1, 5, 10, 15, 20, 30, and 60 min were simulated by taking only one of the samples per specific sampling interval (e.g., one per 60 samples for Δ*s* = 30 min). For each Δ*s*, all possible sampling subsets were simulated (e.g., 60 possible subsets for Δ*s* = 30 min). The TPs resulting from these subset simulations formed an envelope around the baseline profile (naturally, the average of the subset simulations resulted in the baseline profile). The sampling error (denoted as ‘Δ_s,95_’) was then calculated as the maximum (relative) difference of the 97.5th and the 2.5th percentiles of this envelope to the baseline profile. Moreover, the influence of the length of the measuring period was assessed by taking an initial subset of seven days and sequentially adding single measurement days. In this analysis, any non-detects and missing samples (e.g., due to network or power issues) were retained. The results of this analysis for 1800 MHz are shown in Figure 11.

For a target value of Δ_s,95_ below 33% (1.2 dB) in the 1800 MHz band, sampling at a Δ*s* of 60 min would take about 20 additional days for TP_D_ (i.e., roughly four weeks in total), but approximately 14 weeks for TP_W_. When sampling faster, the number of required additional measurements days drops significantly (for example at Δ*s* = 30 min, 0 and 40 days for TP_D_ and TP_W_, respectively). For 2100 MHz only two and three additional weeks are needed, and for the 900 and 2400 MHz bands, all errors were actually below 33%.

The sudden increase in Δ_s,95_ around 110 additional days was due to EMF08 coming online, which showed a larger temporal variation in the 1800 MHz band than the other sensors (Figure 7b and Figure 8b). As the differences were less pronounced for the other bands, the corresponding bumps in Δ_s,95_ were as well.

As can be observed for the 1800 MHz band and to a lesser extent the 2100 MHz band, as the temporal variability increases, more samples (using a longer measurement period and/or shorter sampling time) are needed to mitigate the induced errors. This is becoming increasingly more important with newer telecommunications technologies, such as 4G LTE-Advanced and 5G New Radio (NR), as massive multiple-input-multiple-output (MaMIMO) and beamforming technology massively increase the temporal variability [19].

Furthermore, as observed in Figure 7, Figure 8, Figure 9 and Figure 10, there was quite some inter-sensor variability in the TPs. For spatiotemporal exposure interpolation purposes (see [13]), it would be interesting to cluster sensors into spatial groups with similar temporal behaviour. In this study, hierarchical clustering was applied based on three similarity measures **X**: (a) the correlation between the fixed sensors’ differentiated time series, (b) the correlation between the sensors’ differentiated TP_W_, and (c) the correlation between the differentiated TP_D_. (All three time series were differentiated to remove spurious correlations.) Each measure was then transformed to a distance matrix **1–X**, and the distance matrices were subsequently summed using a one-third weight factor resulting in the weighted distance matrix on which the hierarchical clustering would be based.

Figure 12 shows the range and medians of the three similarity measures per frequency band. It was found that the time series correlation was the lowest for 900 and 2400 MHz, especially for the raw sensor output, i.e., the 30 s electric field strength samples. In these cases, there was also weak negative to moderate positive correlation between the sensor’s TPs. For 1800 and 2100 MHz, the sample correlation was positive (0.1–0.2), with at least some high-correlation outliers. Their TP correlations, however, were moderate to strong, with high outliers.

The high variability of the correlation coefficients and the presence of high values in Figure 12 both point to the existence of pairs of sensors with a higher similarity than others. In the next step, the spatial extent of that similarity was investigated by means of hierarchical clustering using the weighted correlation matrix **r_w_** (Figure 13), which holds the average of the Pearson correlation coefficients of the frequency bands’ differentiated time series of Figure 12 per sensor pair.

Then, clustering was performed using a number of agglomerative hierarchical clustering methods, i.e., ‘single’, ‘average’, ‘complete’, ‘ward’, and ‘weighted’. For each method, the number of clusters was based on the silhouette score [20]. Finally, each sensor was assigned to the cluster it was most often allocated to when using the various methods. The three resulting clusters are shown in Figure 14.

When using temporal profiles at higher resolution in the correlation analysis (e.g., Δ*t* = 30 min instead of 60 min), the resulting correlation coefficients were lower, but the clustering remained the same.

Finally, the bump in Figure 11 shows that a single sensor node can significantly change the overall temporal profile of the area under study, which may in turn indicate that the initial sensor node density was inadequate. The correlation between the fixed sensors’ (differentiated) time series (Figure 13) can also be used as a metric to define adequate sensor density, i.e., such that the maximum minimum (maximin) distance between two sensors is small enough for their measurements to be fairly well correlated in time (e.g., *r_w_* > 0.5). In Figure 15, the weighted time series’ correlation coefficients of Figure 13 were plotted as a function of the distance between the pairs of sensor nodes along with an isotonic regression line, i.e., a (in this case) non-increasing fit by minimizing the mean squared error.

Based on the isotonic regression, the correlation reaches 0.5 at a distance of about 100 m (Figure 15). At 90 m, the highest correlation (0.62) was found for two sensor nodes in the same street (EMF09 and EMF13; Figure 13 and Figure 14). However, at closer inter-sensor distances, lower correlation coefficients were found between sensor nodes placed in other streets (e.g., EMF17 and EMF18; Figure 14). From this, it can be concluded that a minimum density on the order of 100 nodes per km^2^ (such that the average minimum distance between nodes is 100 m) would be required, ideally distributed such that at least one sensor is found in any one street.

### 3.3. Identification of Hotspots in Mobile RF-EMF Sensor Network

The four mobile sensors collected per frequency band an average of 442 k samples between 16 November 2018 and 11 December 2019 (390 days), which corresponds to an average temporal coverage of 38% (range: 13–63%). As the four sensors sampled the same general area over a long time, the same distribution in exposure samples was expected. However, Figure 16, in which the distributions of the logarithmic power densities (when using the individual AAs obtained by the in-lab calibration—Appendix A) are plotted, shows that this was not the case. There were large differences in exposure measurements (e.g., EMF03 compared to EMF05) which may be caused by differences in installation on the postal van roof as well additional equipment installed near the sensor nodes and thus a change in effective AA. Unfortunately, this indicates that the free-space calibration of Appendix A is not usable for metal-mounted sensor nodes. As no additional in situ validation measurements were made and a direct comparison to the fixed sensor measurements was also not possible, as there were few mobile measurements in the fixed sensor network area (Figure 3b), the absolute sensor values were considered unviable.

However, the mobile nodes’ raw received power samples could still be used for hotspot detection. For each sensor and frequency band, any sample above the combination’s 95th percentile was given the value ‘1’, and below the 95th percentile the value ‘0’. The average value per snapped location (Figure 4)—denoted as the ‘hotspot probability’—in the 95% area is shown in Figure 17. Since these are relative hotspots, with no information on the absolute exposure value, this map could be used by regulatory bodies to identify locations in a relatively wide geographical area (here, roughly 6 km by 10 km) where an accurate RF-EMF exposure assessment would be more valuable compared to other locations with a lower expected exposure. For example, in Figure 17, a high hotspot probability is observed directly next to EMF09 and EMF13 (compare Figure 17 to Figure 2a), and relatively higher electric field values were measured by these fixed sensors (Table 2).

## 4. Discussion

A novel low-cost sensor node was designed to measure the radiofrequency (RF) electromagnetic field (EMF) strength in four predetermined frequency bands. These bands were selected based on their use by mobile telecommunications networks in Belgium. However, the sensor design is modular and other frequency bands (e.g., 800 MHz and 2600 MHz for 4G LTE, and 700 MHz, 1400 MHz, or 3500 MHz for 5G NR) can be considered according to the availability of an antenna and a Surface Acoustic Wave (SAW) filter. Since the developed sensor is mains powered, a high sample rate can be achieved (e.g., 1 sample per s). The raw samples can be collected in a central database over a wired internet connection or transmitted over-the-air using e.g., a long-range wide area network (LoRaWAN).

In this study, ten RF sensor nodes were installed on building façades in an urban environment in the city centre of Antwerp, Belgium, with an area of 0.11 km^2^. Between 146 thousand and 1.307 million samples (per 30 s) were collected per sensor and per frequency band over 475 days between January 2019 and May 2020.

As in other studies, e.g., [12], the fixed nodes were placed out of immediate reach for practical and/or security reasons—in this case, at different heights between 4.0–7.5 m. However, the influence of the node height on the exposure assessment at street level (1.5 m) is unclear. Based on the validation measurements with a spectrum analyser (Appendix B), the effect seemed to go both ways (sometimes underestimation, sometimes overestimation) and can be large (between −7 and +9 dB; Figure A2). Hence, it is advised to perform additional spectrum analyser measurements at street level (1.5 m) below every RF sensor and add the difference as a calibration factor to the sensor’s measurements. An additional motivation for this is that the antenna aperture (AA) determined in a lab environment (anechoic chamber) may change when the sensor is placed in situ against a wall of varying construction build-up.

Based on the temporal profiles, the overall diurnal exposure variability in this part of Antwerp compared to Santander [13] was lower in the 900 MHz band, higher in the 1800 MHz band, and similar in the 2100 MHz band. The difference in the 1800 MHz band can be attributed to its use by LTE, which was not as prevalent yet during the measurements in Santander [12,13]. In addition, it is possible that the reliance on GSM (using the 900 MHz band) for voice calls has diminished in favour of calls using voice over internet protocol (VoIP) or LTE (VoLTE), or that there is a regional difference.

Furthermore, aside from a difference in relative exposure between hours of the day, there were also clear differences between days of the week. Figure 6 and Figure 7 emphasize the importance of temporal profiles to rescale measurements taken at different times of the week to properly compare exposure values or to assess the realistic worst-case values.

The high temporal resolution of the fixed sensor network was exploited to assess the potentially induced errors by having to resort to lower sample rates (e.g., in [13]). Furthermore, an adequate sensor density was established, i.e., on the order of 100 sensors per km^2^, for which the average minimum distance between sensors in the area under study is 100 m. However, to obtain a good coverage, at least one sensor should be installed in each street. As the scope in this study (number of fixed sensors and area covered) was limited, more research, with more sensors in a larger area, is needed to determine an optimal sensor density. In the SmartSantander platform, the sensor density was 64 per km^2^ [12,13], which is slightly too low, but a correlation analysis such as in Figure 12, Figure 13 and Figure 15 will be the subject of future research in order to validate the results of this study.

Although a vast number of samples was collected, the area covered by the fixed sensors was small and the inter-sensor variability in RF-EMF exposure was quite low (Table 2 and Table A1). For this study, the number of sensors, as well as the number of suitable deployment locations (with gateways) were limited. In the future, the RF sensors should not be dependent solely on wired internet access, but should use wireless networks, such as mesh networks, LoRaWAN, or Narrowband IoT (NB-IoT). If data rates are limited, longer sampling times can be used (Figure 11). Lastly, if there is no access to the mains, the RF sensor can be battery powered (possibly in combination with solar power). However, in this case, a trade-off between sampling time and measuring time (i.e., the number of days the device can last) can be based on Figure 11.

The fixed sensor nodes were clustered in groups based on the weighted correlation between the (differentiated) raw samples’ time series, diurnal temporal profiles, and weekly temporal profiles. The clusters formed geographically continuous areas that exhibited distinct temporal variations in RF exposure. Hence, this clustering should improve the spatiotemporal interpolation when compared to the area-wide temporal profiles used in [13], but that hypothesis could not be tested in this study because of the limited number of sensors and the limited size of the area. However, it will be tested on the SmartSantander platform.

In addition to the fixed nodes, four RF sensor nodes were installed on the roof of postal vans. The postal vans did not have a fixed schedule or route, nor a fixed speed. Furthermore, the effective capture and transmission of sampling data depended on the vehicle’s driver to plug in the node. Between November 2018 and December 2019 (390 days), between 120 thousand and 720 thousand samples (per 30 s) were collected per mobile sensor and per frequency band. Moreover, the mobile measurements were performed in an area roughly 6% the size of Belgium (Figure 16a), with 95% of the samples were taken in an area of 6.6 km by 10.5 km (3.8% of the total area).

The sampling time of the mobile sensors was 30 s. At a speed of 30–50 km/h (i.e., the legal speed inside the city centre), this meant that the distance between successive samples could reach about 250–420 m. For comparison, the maximum distance between samples in [10] was 2.3 m. However, the sample rate was limited due to the LoRaWAN data rate. Internally, samples were collected every 5 ms, i.e., every 4–7 cm at the mentioned speeds. Thus, in the future, higher sample rates, which are possible with the described sensor, or other means of mobility, such as bicycle couriers (perhaps with an electric bicycle which could also power the sensor) or personal exposimeters measurements [21], could be recommended.

The mobile sensor nodes were not validated in situ, and there were almost no mobile measurements in the area covered by the fixed nodes (Figure 16b). In addition, it was deemed probable that the in-lab free-space calibration of the car-mounted sensors would not yield an accurate exposure evaluation [9]. Hence, the sensors’ received power samples were not converted to electric field strengths or power density values. Instead, the mobile RF nodes were used to identify potential hotspots of RF-EMF exposure (in the considered bands) by pinpointing the received power samples above the 95th percentile. This resulting map could be used by regulatory bodies to identify locations in a relatively wide geographical area where an accurate RF-EMF exposure assessment would be valuable. In addition, mobile network providers could be interested in this data for coverage monitoring, although no distinction can be made between technologies and specific frequency bands used by individual operators.

When correctly calibrated, the mobile measurement data set can be used to assess the outdoor RF-EMF exposure in the entire city centre of Antwerp (Figure 3b) by applying spatial interpolation techniques such as kriging [10,18,22] or models based on artificial neural networks [23]. Moreover, spatiotemporal assessment can be obtained by scaling the samples taken at any time instance to any other time instance via the temporal profiles obtained through the fixed sensor network [13]. This combination of fixed and mobile data should be further explored.

Based on Figure 11, the case could further be made for nomadic sensor nodes. Battery-powered sensors are less location dependent and can be moved to another location after an initial deployment. Moreover, during each move, the battery can be changed. The sampling time and the measuring time could be based on the battery specifications and Figure 11. A nomadic sensor network would be an interesting hybrid of mobile and fixed sensor networks: nomadic nodes could cover a large area in a relatively limited time while keeping a high temporal resolution (if high sample rates can be maintained). At each new location, a validation measurement should be performed (Appendix B). New locations could be selected based on the hitherto nomadic data set [18] or based on mobile measurements in the area (Figure 17).

Finally, this study focused on frequency bands with downlink telecommunications signals. Due to the use of frequency division duplexing (FDD), these bands do not include any signals from user devices, i.e., uplink (except for Wi-Fi). In future sensor node designs, uplink exposure can be measured by adding other frequency bands, or, in case of 5G NR or 4G LTE, specific frequency bands with time division duplexing (TDD).

## 5. Conclusions

In this study, a low-cost RF sensor node was designed that is essential for a large-scale spatiotemporal assessment of human exposure to radiofrequency electromagnetic fields (RF-EMF). The mains-powered sensor node, built with off-the-shelf components, can sample four RF bands used by mobile telecommunications networks per second. An RF-EMF exposure sensing network was deployed in the city of Antwerp, Belgium, comprising ten RF sensor nodes installed at fixed locations and four sensors placed on the roof of postal vans with a-priori unknown schedules. Between November 2018 and May 2020, slightly less than 10 million samples (at one sample per 30 s) were collected by the combined networks. The high variability in the measured exposure observed over time further underlined the need for temporal monitoring. In the area covered by the fixed sensors, the electric field levels per frequency band ranged from below sensitivity (0.02 V/m) to 0.75 V/m. As often observed, overall, the 900 MHz frequency band resulted in the dominant contribution to the total exposure. The high temporal resolution of the fixed sensors was then exploited to assess the potential errors induced when sampling at lower speeds and measuring for shorter periods of time. Moreover, an adequate resolution, which preserves short-term variations, for diurnal and weekly temporal profiles was found to be 30–60 min, which aligns with the ICNIRP guidelines for the assessment of environmental RF-EMF exposure of the general public. Then, based on the spatial correlation between the sensor nodes’ time series, an adequate density of 100 nodes per km^2^ was deduced. Whereas the fixed EMF exposure sensing network enabled accurate spatiotemporal exposure assessment on a local scale (0.1 km^2^), the mobile measurements were used to identify potential hotspots of RF-EMF exposure in a wide geographical area (on the order of 10–100 km^2^), which is of high interest to regulatory bodies. In summary, this paper offers invaluable insights applicable to future designs and deployments of distributed RF-EMF sensor networks: from the initial deployment (e.g., the sensor placement and density) to the trade-off between the sampling rate and the total measuring period (e.g., in the case of battery-powered sensors). Finally, various techniques are proposed to improve the spatiotemporal assessment of RF-EMF exposure in large geographical areas, such as nomadic sensors, optimized temporal profiles, time-series clustering, and spatiotemporal interpolation techniques. In the future, these techniques will be verified in a larger RF-EMF sensing network.

## Figures and Tables

**Figure 1 sensors-22-01715-f001:**
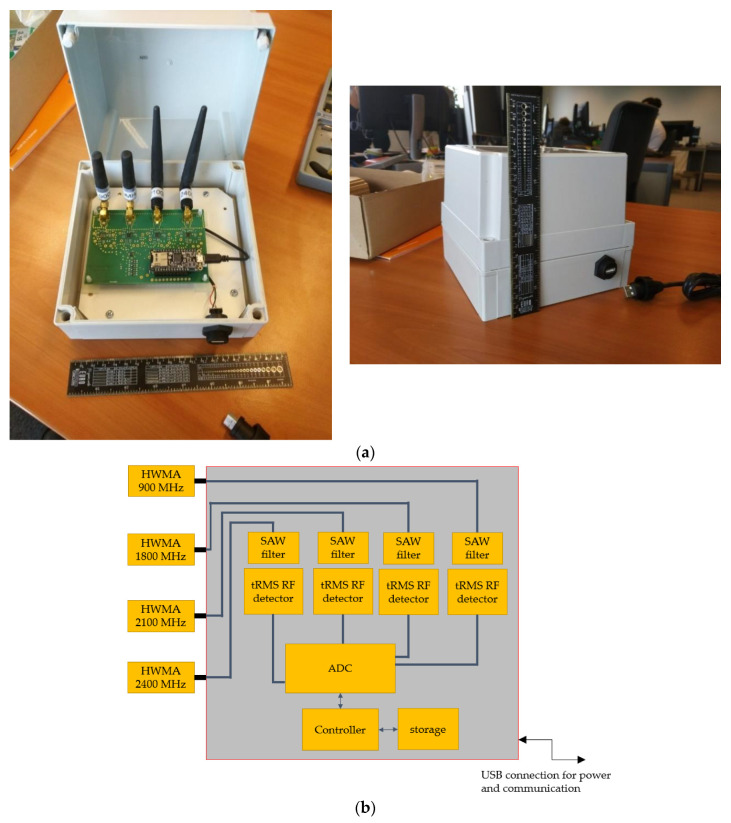
(**a**) The radiofrequency (RF) electromagnetic field (EMF) sensor in its casing. The black stubs are RF-EMF antennas for different frequency bands (Table 1), the green printed circuit board (PCB) is a custom RF-EMF measurement system, the blue PCB is a commercially available communication module with a microcontroller, and the white casing is a custom polyvinylchloride casing; (**b**) Block diagram of the RF EMF sensor. HWMA, half-wavelength monopole antenna; SAW, surface acoustic wave; tRMS, true root-mean-square; ADC, analogue-to-digital convertor; USB, universal serial bus.

**Figure 2 sensors-22-01715-f002:**
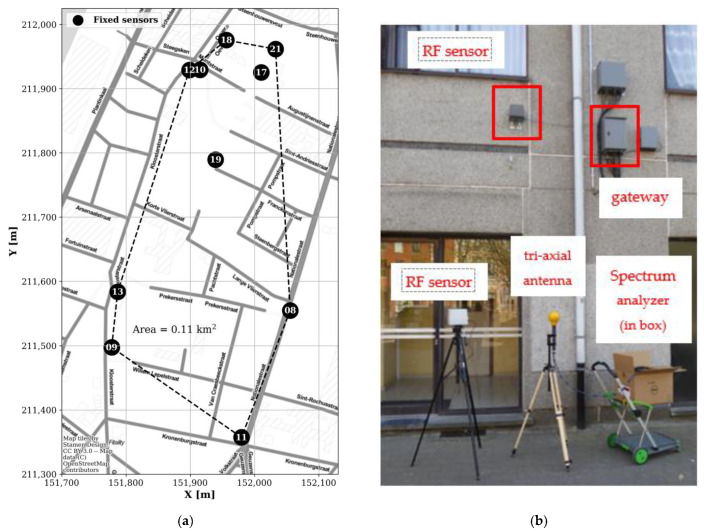
(**a**) Locations of the fixed RF-EMF sensor nodes (labelled with their ID number) in an area of 0.11 km^2^ in the IMEC City of Things SmartZone in Antwerp, Belgium (coordinate system: Belgian Lambert 1972). (**b**) Fixed RF-EMF sensor node and gateway mounted on a building façade, and the in situ validation setup (see Appendix B) consisting of an additional RF sensor node (EMF20) and a spectrum analyser and tri-axial antenna at ground level.

**Figure 3 sensors-22-01715-f003:**
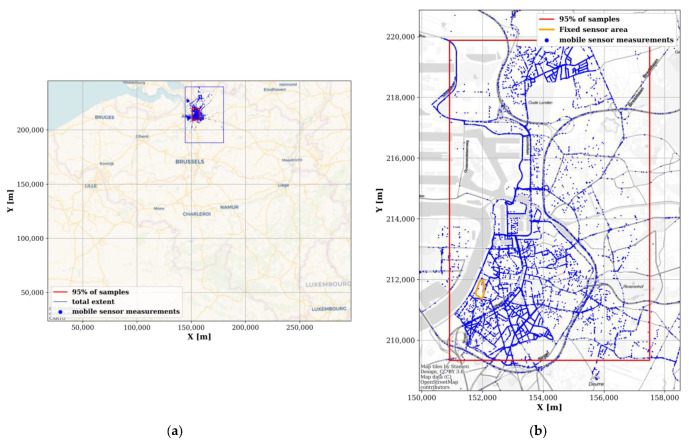
Extent of the mobile RF sensor measurements (blue dots). (**a**) Area covered by the mobile RF sensors (blue rectangle) in comparison to Belgium. (**b**) Zoom of the covered area, with sample locations after processing the GPS data as shown in Figure 4 [16]. The area bounded by the red rectangle contains 95% of the measurement locations. The area bounded by the orange polygon is the area bounded by the fixed sensor network (Figure 2a) (coordinate system: Belgian Lambert 1972).

**Figure 4 sensors-22-01715-f004:**
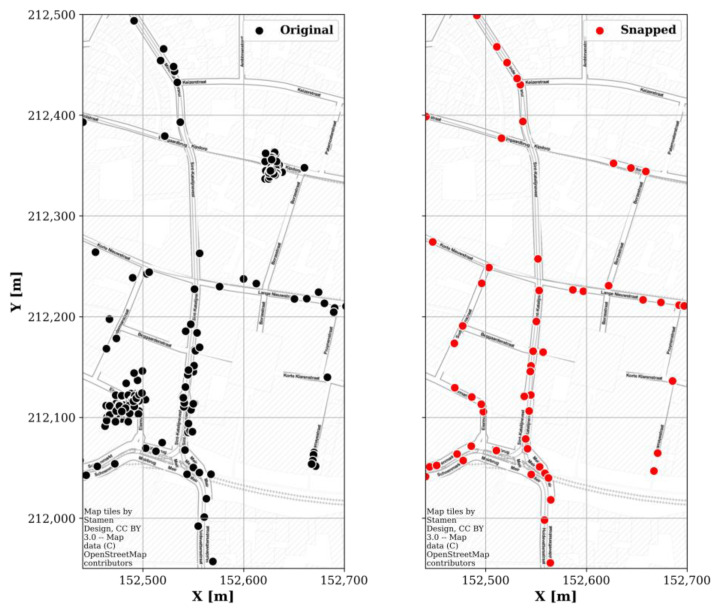
Original GPS coordinates of mobile sensor samples (**left**) and processed ‘snapped’ coordinates (**right**) [16] (coordinate system: Belgian Lambert 1972).

**Figure 5 sensors-22-01715-f005:**
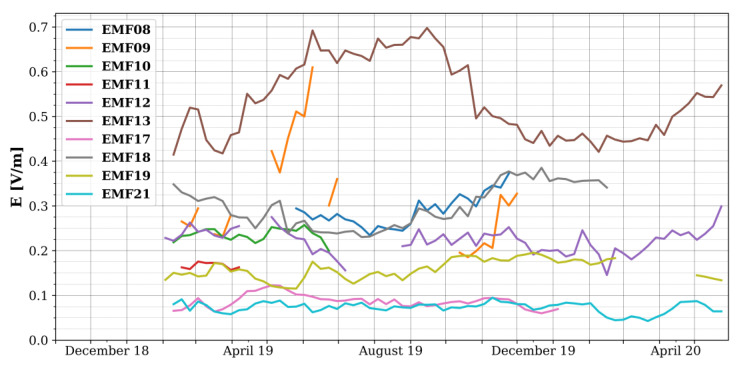
Weekly average electric field strength as measured by the fixed RF-EMF sensors in the 900 MHz frequency band. EMF08 (dark blue) and EMF09 (orange) were not validated in situ.

**Figure 6 sensors-22-01715-f006:**
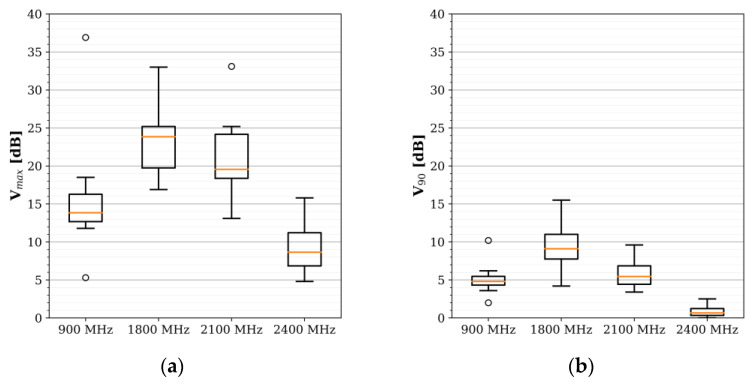
(**a**) Maximum and (**b**) 90% temporal variability of the signals. The maximum variability is calculated as the logarithmic difference of the maximum and minimum electric field samples, 20log_10_(E_max_/E_min_), whereas 90% equals the difference between the 95th and 5th percentiles [13].

**Figure 7 sensors-22-01715-f007:**
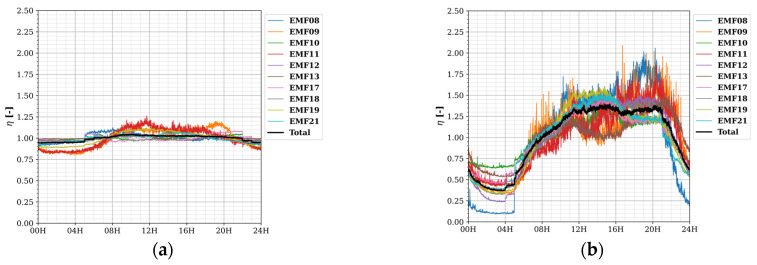
Diurnal temporal profiles TP_D_ with resolution Δ*t* of 30 s for each fixed RF sensor (and the area total) and for different frequencies (**a**) 900 MHz; (**b**) 1800 MHz; (**c**) 2100 MHz; and (**d**) 2400 MHz. Here, *η* is the average power density at a certain time of the day normalized to the daily average.

**Figure 8 sensors-22-01715-f008:**
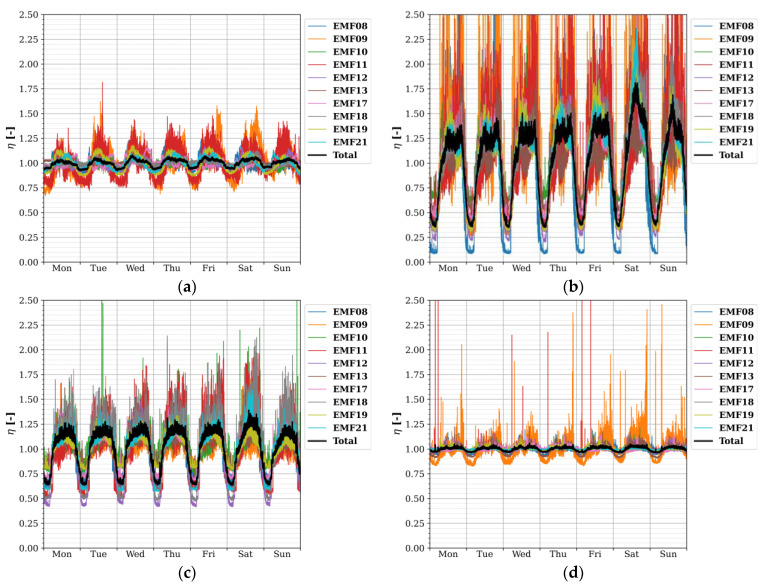
Weekly temporal profiles TP_W_ with resolution Δt of 30 s for each fixed RF sensor (and the area total) and for different frequencies (**a**) 900 MHz; (**b**) 1800 MHz; (**c**) 2100 MHz; and (**d**) 2400 MHz. Here, *η* is the average power density at a certain time of the week normalized to the weekly average. Some data for 1800 and 2400 MHz were censored to allow for visual comparison of the different frequencies (*η* values reached up to 5 for 1800 MHz).

**Figure 9 sensors-22-01715-f009:**
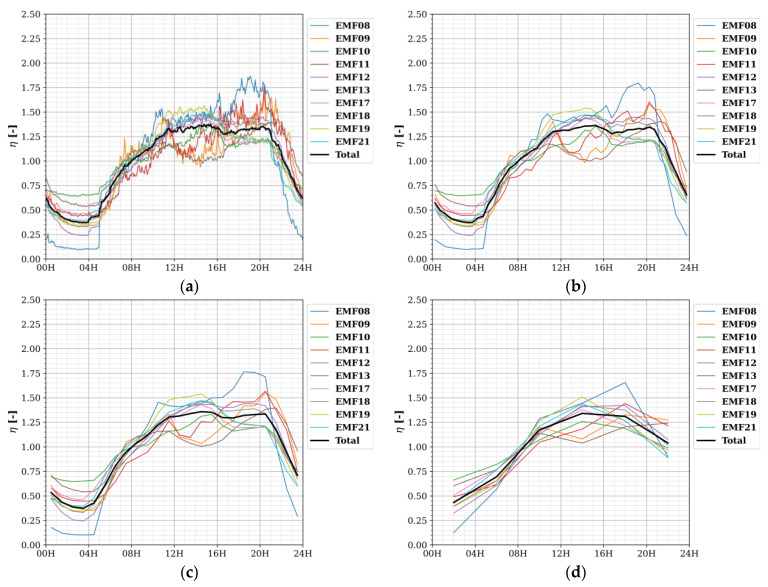
Diurnal temporal profiles for 1800 MHz with (**a**) Δ*t* = 5 min, (**b**) Δ*t* = 30 min, (**c**) Δ*t* = 1 h, and (**d**) Δ*t* = 4 h.

**Figure 10 sensors-22-01715-f010:**
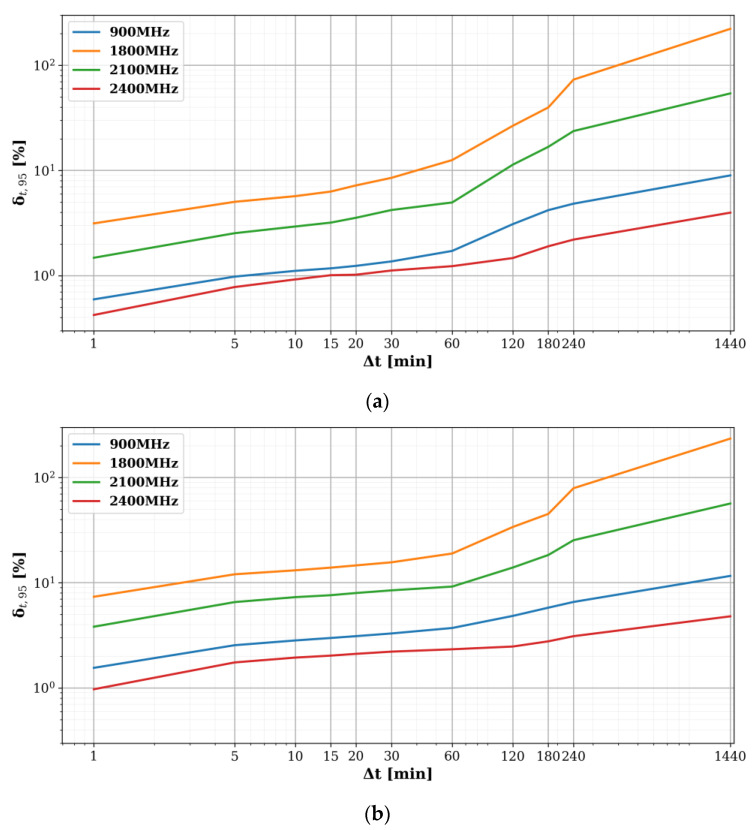
Sensor-averaged 95th percentiles of the relative errors Δ_t,95_ when comparing (**a**) diurnal (TP_D_) and (**b**) weekly temporal profiles (TP_W_) with different resolutions Δt to the baseline TP with Δ*t* = 30 s (Figure 7 and Figure 8). Δ*t* = 1440 min (=24 h) indicates the induced error when using only daily averages.

**Figure 11 sensors-22-01715-f011:**
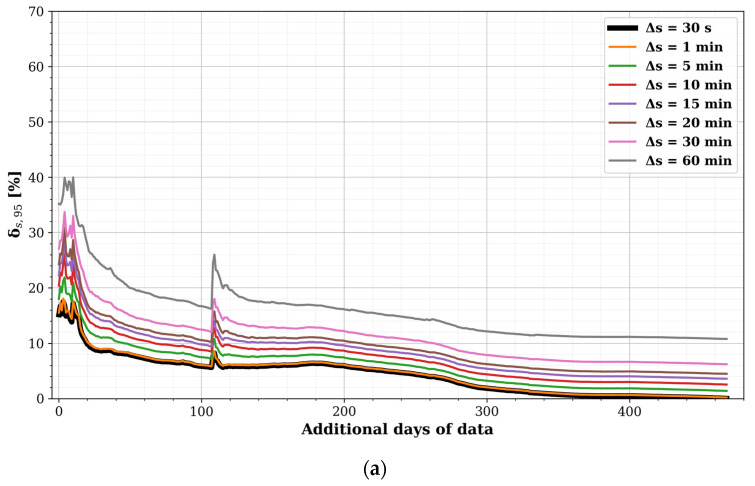
Sensor-averaged sampling errors Δ_s,95_ induced in (**a**) TP_D_ and (**b**) TP_W_ for 1800 MHz with a resolution of Δ*t* = 60 min, when using different sampling times Δ*s* between 1 and 60 min, as a function of the additional days of measurement data after an initial 7-day period.

**Figure 12 sensors-22-01715-f012:**
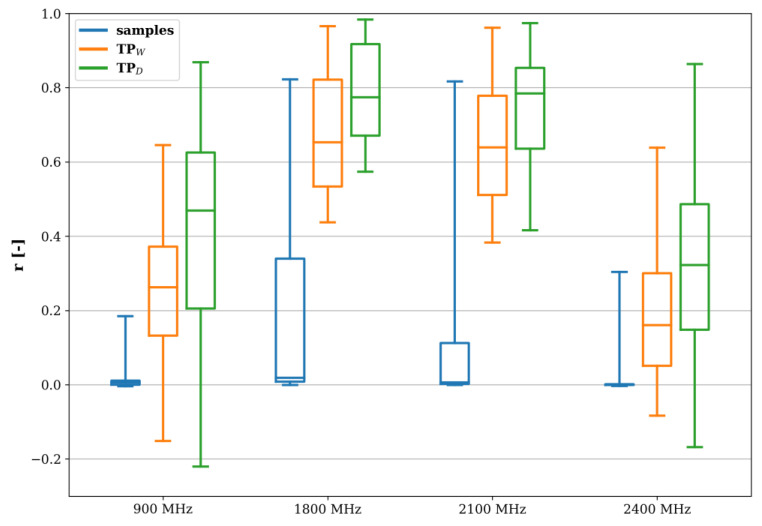
Boxplots showing the median and the range of the Pearson correlation coefficients per frequency band between the differentiated time series measured by the fixed RF-EMF sensors denoted as ‘samples’, ‘TP_D_’ (daily temporal profile), and ‘TP_W_’ (weekly temporal profile).

**Figure 13 sensors-22-01715-f013:**
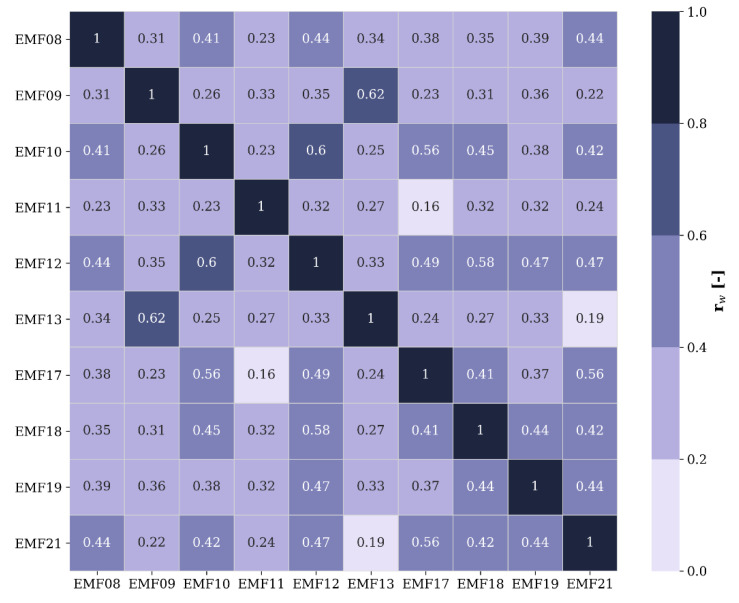
Weighted time series correlation matrix.

**Figure 14 sensors-22-01715-f014:**
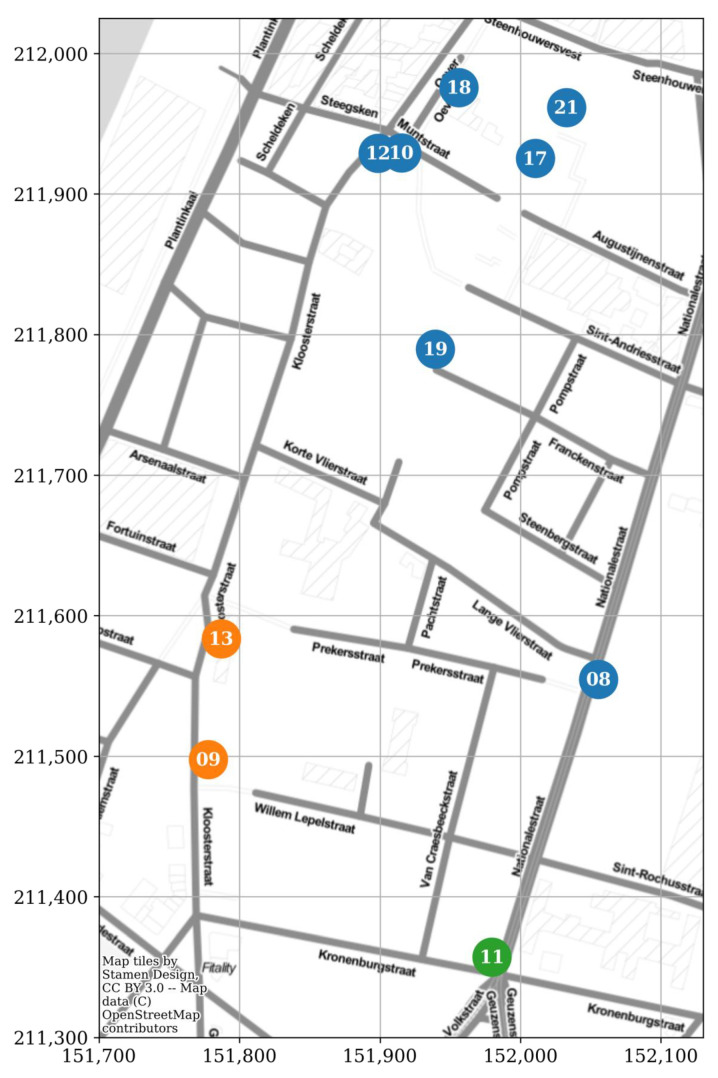
Clustering of the fixed RF-EMF sensors (labelled with their ID number) based on their temporal correlation (coordinate system: Lambert 1972).

**Figure 15 sensors-22-01715-f015:**
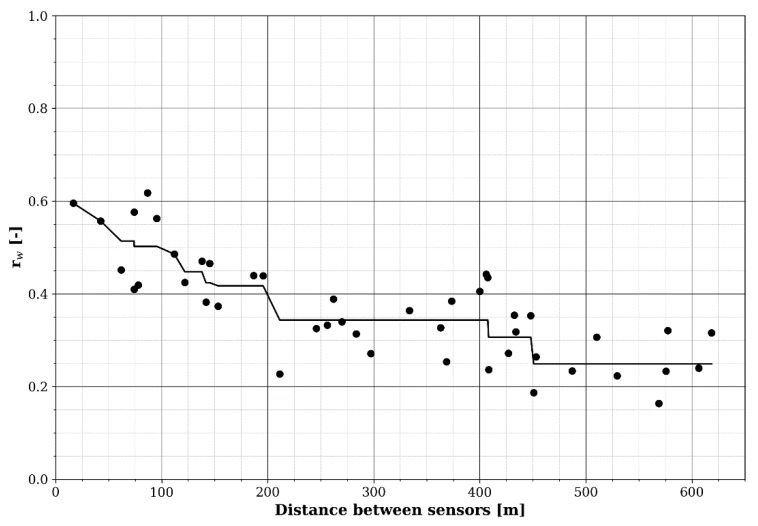
Weighted correlation between fixed RF-EMF sensor pairs as a function of the distance between them. The black line is an isotonic regression.

**Figure 16 sensors-22-01715-f016:**
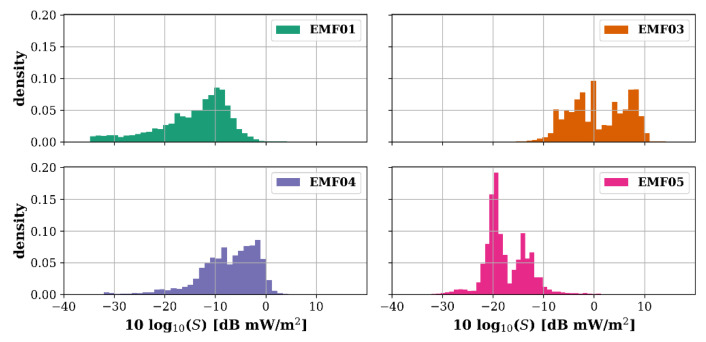
Histograms of the logarithmic power density measured by the mobile RF sensors when applying the free-space calibration antenna apertures.

**Figure 17 sensors-22-01715-f017:**
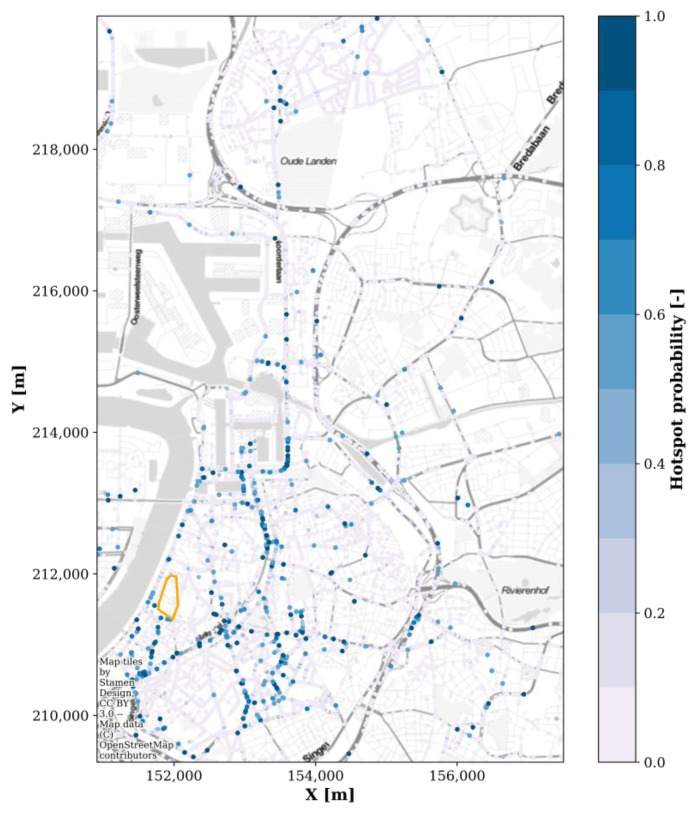
Probability of hotspots in the area covered by 95% of the mobile RF sensor samples. The orange delineated area is the area covered by the fixed RF sensor network (coordinate system: Belgian Lambert 1972).

**Table 1 sensors-22-01715-t001:** Specifications of the radiofrequency (RF) electromagnetic field (EMF) sensor.

Parameter	Value
Frequency Range	925–2484 MHz
Frequency band 1	‘900 MHz’: 925–960 MHz
Frequency band 2	‘1800 MHz’: 1805–1880 MHz
Frequency band 3	‘2100 MHz’: 2110–2170 MHz
Frequency band 4	‘2400 MHz’: 2400–2484 MHz
Sensitivity level	0.02 V/m
Dimensions (L × W × H)	18 × 18 × 15 cm^3^
Dynamic range	70 dB
Supply voltage	5 VDC USB power
Power consumption	Max. 150 mA (0.75 W)
Output sampling time Δ*s*	1000 ms
Internal sampling time	5 ms
Output format fixed nodes	ASCII UART serial output at 9600 baud
Output format mobile nodes	I^2^C ASCII output

VDC, volts of direct current; UART, universal asynchronous receiver–transmitter; I^2^C, Inter-Integrated Circuit; ASCII, American Standard Code for Information Interchange.

**Table 2 sensors-22-01715-t002:** Number of 30 s samples (and corresponding temporal coverage or relative on-time) and the 99% interval (0.5th to 99.5th percentiles) of the electric field values *E*_30s_ (in V/m) measured by the ten fixed RF sensor nodes in the IMEC City-of-Things network in Antwerp, Belgium. For each validated sensor (all nodes except EMF08 and EMF09), the individual offsets compared to the SRM validation setup were added to the measurements to better predict the exposure at ground level (see Appendix B).

Node ID	Number of 30 s Samples	Temporal Coverage (%)	99% Interval of *E*_30s_ (V/m)			
			900 MHz	1800 MHz	2100 MHz	2400 MHz
EMF08 *	508,398	37.09	0.19–0.40	0.05–0.62	0.10–0.20	0.02–0.03
EMF09 *	280,823	20.48	0.13–0.60	0.06–0.32	0.08–0.19	0.02–0.03
EMF10	323,434	23.59	0.20–0.28	0.11–0.25	0.07–0.17	0.03–0.04
EMF11	145,647	10.62	0.12–0.22	0.06–0.31	0.06–0.17	0.03–0.03
EMF12	1,127,490	82.25	0.12–0.31	0.03–0.18	0.06–0.23	0.03–0.04
EMF13	1,307,392	95.37	0.35–0.75	0.17–0.63	0.07–0.13	0.03–0.04
EMF17	922,189	67.27	0.05–0.13	0.03–0.11	0.02–0.05	0.03–0.06
EMF18	1,047,889	76.44	0.19–0.41	0.03–0.21	0.04–0.19	0.03–0.03
EMF19	1,105,401	80.63	0.10–0.22	0.02–0.11	0.03–0.07	0.03–0.03
EMF21	1,321,186	96.38	0.03–0.10	0.03–0.12	0.02–0.06	0.03–0.03
**Avg.**	**808,985**	**59.01**	**0.17–0.39**	**0.08–0.34**	**0.06–0.16**	**0.03–0.04**

* Not validated in situ.

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
