# Peer review of "Lessons Learned from a Distributed RF-EMF Sensor Network"

_sensors, 2022, doi:10.3390/s22051715_

Round 1

Reviewer 1 Report

The paper entitled "Lessons Learned from a Distributed RF-EMF Sensor Network” reports, as the title indicates, describes a study of a low-cost RF-EMF sensor node capable of monitoring four frequency bands used by wireless telecommunications with good temporal resolution, and its application in a small-scale distributed sensor network consisting of both fixed (on building façades) and mobile sensor nodes (on postal vans), and the subsequent analysis of over a year of data acquisition. 

The results are  technically valid and relevant.

 I recommend the paper to be published in Sensors, although there are some comments, as follows:

In section 2 - A circuit or block diagram may have been helpful in better understanding the device. Therefore, the understanding of the project design was somewhat compromised.

In line 276 - format error

Author Response

Dear reviewer

Thank you. We fixed the formating error(s) and added a block diagram of our device to Figure 1.

Reviewer 2 Report

This paper describes a comprehensive study comprising the design of a low-cost RF-EMF sensor node capable of monitoring four frequency bands used by wireless telecommunications with an unparalleled temporal resolution. This paper has a very clear description of the corresponding work, I suggest to accept it, but the following issues need to be revised.
(1)The paragraphs are divided too frequently, making me feel less like a research paper, but more like a report, you need to organize well.
(2) You should show the detailed circuit configuration of radiofrequency (RF) electromagnetic field (EMF) sensor.
(3) There are too many diagrams in this paper, you should show the main information and group similar diagrams together instead of listing them all.
(4) how about the anti-interference ability of this RF EMF sensor in strong electromagnetic fields?
(5) you mentioned that a low-cost RF sensor node was designed, but the description on the low-cost is not clear.

Author Response

Dear reviewer

Thank you for appreciating our paper. Please find below our answers to your comments:

(1) The paragraphs are divided too frequently, making me feel less like a research paper, but more like a report, you need to organize well.

  • We reduced the number of subsections and paragraphs. The results section now starts with the following paragraph:

“The results of this study are divided into three parts. In the first part, the temporal behaviour of the RF-EMF measured by the sensors in the fixed network is analysed. Based on this, the second part explores in detail three design parameters of a future sensor network: the sampling speed of the sensor, the temporal resolution of the output data, and the spatial density of the sensor network. Finally, in the last part, the search for RF-EMF exposure hotspots in an extensive geographical area using a mobile sensor network is examined.”

(2) You should show the detailed circuit configuration of radiofrequency (RF) electromagnetic field (EMF) sensor.

  • We added a block diagram in Figure 1, as also suggested by another reviewer.

(3) There are too many diagrams in this paper, you should show the main information and group similar diagrams together instead of listing them all.

  • We understand that there is a lot of information in this paper, and even added another diagram to Figure 1. However, it felt necessary for us to include all of the lessons we learned from this network, as opposed to split the results into multiple smaller less-comprehensive papers. Moreover, moving sections to the Appendix would disrupt the flow of the paper more than improve it.

(4) how about the anti-interference ability of this RF EMF sensor in strong electromagnetic fields?

  • We added the following in Section 2.1: “Finally, it is worth mentioning that outside the pass bands (Table 1), the attenuation of the SAW filters is typically between 17 dB and 24 dB relative to the passband insertion loss.”

(5) you mentioned that a low-cost RF sensor node was designed, but the description on the low-cost is not clear.

  • All of the components in our RF sensor are off the shelf – nothing was specifically designed – to minimize the costs. We mentioned this in the paper, and we added now that “all the components are off the shelf and no specific component design was done.” in Section 2.1.

Reviewer 3 Report

The authors have presented a very detailed analysis of RF-EMF exposure rates by assessing a contained number of RF-EMF sensor nodes, which featured mixed characters and which were placed in a wide inhomogeneous urban area in order to get a generalizable estimate of the average exposure to EM fields on smart cities.

This study contains deep insights on all the passages required to calibrate the instruments, collect data and extract valuable meanings from them; also, more importantly and really worthy of mention, they reported the peculiar limitations and comprehensible small defects sparsely present in their analysis, such as the unavoidable uncertainty of some measurements due to the delicate dependence of measurement quality on either the chosen detection procedure or to some special sensor nodes, so setting the need for further suggestions at the lowermost levels

They finally conclude their analysis declaring an estimate of 100m for the nodes distance (ideally distributed such that at least one sensor is found in any one street) as an optimal trade-off in order to get a minimal pervasiveness and simultaneously get substantial clues on the RF-EMF exposure across the population.

I personally consider this study well prepared, important for the public dissemination of data to get a clear an open knowledge on RF-EMF exposure levels, especially in sight of incoming 5G and 6G mobile technologies, which will surely deeply affect our lives and alterate (by rising them) the current EM levels. Data have been extensively collected and the instrumentation have been wisely used, with a good care of technical data interpretation and they also underlined which parameters deserved better inspections and future debates.

I therefore believe that this document is worthy of publication.

If searching for defects of this draft to highlight, I could cite its maybe very long extension. Strictly speaking, this draft could appear more suitable to be part of a report of a telecommunications company rather than a classical paper for academic dissemination of scientific progresses, but it is also important to note that the analysis here presented requires extensive explanations and dissertations, thus leading to partially unavoidable lengths.

Some very tiny defects can be traced on rare typos ("Figure 3n" in line 166 maybe stands for "Figure 3b"; line 276 "Error! Reference source not found"), and I assume the authors know how to fix them.

Author Response

Dear reviewer

Thank you for the kind words, they are much appreciated.

We are indeed aware of the length of this paper. However, it felt necessary to add all of the things we learned from this network, as opposed to split it into multiple smaller less comprehensive papers. Based on the suggestion of another reviewer, we reduced the number of subsections and paragraphs. The results section now starts with the following paragraph:

“The results of this study are divided into three parts. In the first part, the temporal behaviour of the RF-EMF measured by the sensors in the fixed network is analysed. Based on this, the second part explores in detail three design parameters of a future sensor network: the sampling speed of the sensor, the temporal resolution of the output data, and the spatial density of the sensor network. Finally, in the last part, the search for RF-EMF exposure hotspots in an extensive geographical area using a mobile sensor network is examined.”

We also fixed the typo and formating error. Thanks for pointing them out.

Reviewer 4 Report

The paper is well organized and provides useful practical data. The statistical treatment of the measured samples is well designed and provides an insightful picture of the EM frequency profile in a densely populated area and its surroundings. Maybe, a suggestion of how these measured data would influence further development of the existing networks and some recommendations for the companies in what concerns new network standard implementation may complete the proposed valuable study. The comparison made with a similar study (SmartSantander IoT platform) is also beneficial in the context of better assessing the presented datasets.

Author Response

Dear reviewer

Thank you for appreciating our paper.

We believe that sensor network data are of interest to policy makers and the general public and may be less suitable for the development of mobile networks. We added the following sentence to the Discussion: “In addition, mobile network providers could be interested in this data for coverage monitoring, although no distinction can be made between technologies and specific frequency bands used by individual operators.” For the development of sensor networks, our recommendations are in the Conclusions.

Throughout the paper, the results were compared to those in Santander (each time we refer to references [12] and [13]). However, we added the following sentences in the Discussion :

  • “In the SmartSantander platform, the sensor density was 64 per km2 [12,13], which is slightly too low, but a correlation analysis such as in Figures 12, 13, and 15 will be the subject of future research in order to validate the results of this study.”
  • “Hence, this clustering should improve the spatiotemporal interpolation when com-pared to the area-wide temporal profiles used in [13], but that hypothesis could not be tested in this study because of the limited number of sensors and the limited size of the area. However, it will be tested on the SmartSantander platform.